# NARX Technique to Predict Torque in Internal Combustion Engines

Federico Ricci [ID], Luca Petrucci *[ID], Francesco Mariani [ID] and Carlo Nazareno Grimaldi [ID]

Engineering Department, University of Perugia, Via Goffredo Duranti, 93, 06125 Perugia, Italy; federico.ricci@unipg.it (F.R.); francesco.mariani@unipg.it (F.M.); carlo.grimaldi@unipg.it (C.N.G.)
* Correspondence: luca.petrucci89@gmail.com

**Abstract:** To carry out increasingly sophisticated checks, which comply with international regulations and stringent constraints, on-board computational systems are called upon to manipulate a growing number of variables, provided by an ever-increasing number of real and virtual sensors. The optimization phase of an ICE passes through the control of these numerous variables, which often exhibit rapidly changing trends over time. On the one hand, the amount of data to be processed, with narrow cyclical frequencies, entails ever more powerful computational equipment. On the other hand, computational strategies and techniques are required which allow actuation times that are useful for timely and optimized control. In the automotive industry, the 'machine learning' approach is becoming one the most used approaches to perform forecasting activities with reduced computational effort, due to both its cost-effectiveness and its simple and compact structure. In the present work, the nonlinear dynamic system we address is related to the torque estimation of an ICE through a nonlinear autoregressive with exogenous inputs (NARX) approach. Preliminary activities were performed to optimize the neural network in terms of neurons, hidden layers, and the number of input parameters to be assessed. A Shapley sensitivity analysis allowed quantification of the impact of each variable on the target prediction, and therefore, a reduction in the amount of data to be processed by the architecture. In all cases analyzed, the optimized structure was able to achieve average percentage errors on the target prediction that were always lower than a critical threshold of 10%. In particular, when the dataset was augmented or the analyzed cases merged, the architecture achieved average prediction errors of about 1%, highlighting its remarkable ability to reproduce the target with fidelity.

**Keywords:** machine learning; NARX technique; ICE; torque; time-series modeling

## 1. Introduction

The control of internal combustion engines (ICEs) is becoming ever more complex due to both the increasingly stringent regulations on pollutant emissions and customers' requirements for improved performance [1–3]. Because of the increasing engine complexity, the analytical data instruments must manage huge amounts of data from numerous physical sensors during the engine calibration and run-time operations [4,5]. In this way, considerable computational efforts are required to optimize the engine performance, thus leading to a dramatic increase in operating times and costs [6]. Therefore, the main efforts of automotive researchers have aimed to discover advanced technologies capable of effectively monitoring the engine parameters [7–9]. Machine learning (ML) approaches are increasingly proposed in many automotive applications such as virtual sensors [10,11], fault diagnosis systems [12], and performance optimizations [13] for real-time and low-cost hardware implementation and compact configuration [14]. Their capability to forecast parameters employing interpolation-based algorithms of known intermediate values can reduce the number of analyzed operating points, thus leading to notable advantages in terms of memory and computational speed [15–17].

Among the ML approaches, a non-linear autoregressive network with exogenous inputs (NARX) [18,19], i.e., a recurrent dynamic neural network used to model nonlinear

dynamic systems and applied in time series, seems to be a promising method to perform signal analysis inherent to the internal combustion engine. Taghavi et al. [20] compared the capability of a NARX structure to predict the start of combustion of a HCCI (homogeneous charge compression ignition) one-cylinder Ricardo engine with multi-layer perceptron (MLP) and radial basis function (RBF) networks. The NARX architecture showed the best regression coefficient and reduction of computational time. Kitanovic et al. [21] utilized NARX to minimize the fuel consumption of a parallel hydraulic hybrid powertrain system of a transit bus. Quantities like instantaneous vehicle speed, driveshaft torque, hydraulic machine load, and hydro-pneumatic accumulator gas pressure were selected as inputs to the architecture. Fuel consumption decreases and a value of about 80% of the optimally achievable fuel savings can be reached by the NARX approach. Hamid Asgari et al. [22] reported the ability of open-loop and closed-loop NARX models to predict the dynamic behavior of a single-shaft gas turbine over different operational ranges. Salehi et al. [23] showed the effectiveness of a NARX structure to model a fuel flow control system of a turboshaft gas turbine engine.

Within this context, the present work uses the NARX technique for the prediction of the torque delivered by an internal combustion engine (ICE). The neural architecture used was trained and tested on experimental data from physical sensors and an ECU (engine control unit) under different operating conditions, on a port fuel injection (PFI) three-cylinder spark-ignition (SI) engine.

In the first part of the work, preliminary activities were carried out on a specific case to find out the best combination of neurons and hidden layers able to predict a defined target with the lowest errors [24]. A preliminary activity was also performed using the Shapley method, which allowed definition of the less influential parameters for the torque prediction [25]. A new neural NARX structure was therefore defined, starting from the reduced dataset, by following the same procedure adopted for the first architecture. Even in this case, the target is to select the best combination of neurons and the number of hidden layers able to predict the defined target with the lowest errors. The performance of the tested architectures was compared and the structure performing best was chosen to predict a series of provided torque signals.

The results showed that the optimized structures were able to reproduce the target torque in all cases analyzed. In particular, the structures operating with reduced inputs exhibited higher performance with smoother fluctuations, consistent behavior, and average percentage errors always lower than a critical threshold of 10%. When the initial dataset was augmented or the analyzed cases randomly merged, the best architecture achieved average prediction errors up to 1%, and in any case always lower than 4%.

## 2. Experimental Setup

Tests were carried out on a 999 cc 3-cylinder engine SMART W451T turbocharged with 16 valves and pent-roof combustion chambers (Table 1). The maximum power of 84 CV was produced at 5250 rpm and the maximum torque was equal to 120 Nm at 3250 rpm. The internal cylinder bore was 72 mm while the piston stroke was 81.8 mm. The compression ratio was equal to 10:1. The engine was designed to operate with port fuel injection (PFI) with the igniter, i.e., spark, centrally located. Standard European market gasoline (E5, with RON = 95 and MON = 85) was injected at a fixed absolute pressure of 4.2 bar using port fuel injectors (Mitsubishi 1465A337). A Borghi & Saveri eddy current brake dynamometer of 600 CV was coupled with the crankshaft to ensure the engine speed in the firing condition (Figure 1). A Vascat electric motor of 66.2 kW was added in a tandem configuration to control the engine speed both in motored and firing conditions. All the engine parameters, such as, for instance, ignition timing, injector energizing, turbocharged rate, and so on, were controlled using an EFI EURO-4 engine control unit (ECU). The signals from thermocouples TCK and PTX 1000 pressure sensors were respectively acquired by data acquisition system modules of National Instruments type CFP-CB-3 and type CFP-AI-110. The indicated analysis was performed through a Kistler Kibox combustion analysis system (maximum

temporal resolution of 0.1 CAD) that acquired the pressure signals from the piezoresistive sensors (Kistler 4624A) placed in the intake and exhaust ports, the in-cylinder pressure of the piezoelectric sensor (Kistler 5018) placed on a side of the combustion chamber beside the flywheel, the ignition signal from the ECU, and the absolute crank angular position measured by an optical encoder (AVL 365C). Due to structural and mechanical limitations, only the combustion chamber beside the flywheel was fitted with a piezoelectric sensor, which was used to determine the indicated mean effective pressure (IMEP). The determination of the fuel consumption was realized using a dynamic fuel meter AVL 733S. A torquemeter placed close to the engine crankshaft was used to determine the torque delivered by the engine. The speed of the turbocharger was measured by a rotational speed sensor Picoturn Ptcm V1.1. During the engine operations, all the mentioned quantities were recorded by dedicated software provided by Eurins srl called AdaMo Hyper, which allows simultaneous control of speed, torque, and the valve throttle position of the engine both in firing and motored conditions. Figure 1 summarizes the experimental layout.

**Table 1.** Main features of the metal engine.

| Displacement | 999 cc |
|---|---|
| Cylinders | 3 Cyl./4 V per Cyl. |
| Bore | 72 mm |
| Stroke | 81.8 mm |
| Compression ratio | 10:1 |
| Engine configuration | Inline |
| Power | 84 CV at 5250 rpm |
| Torque | 120 Nm at 3250 rpm |

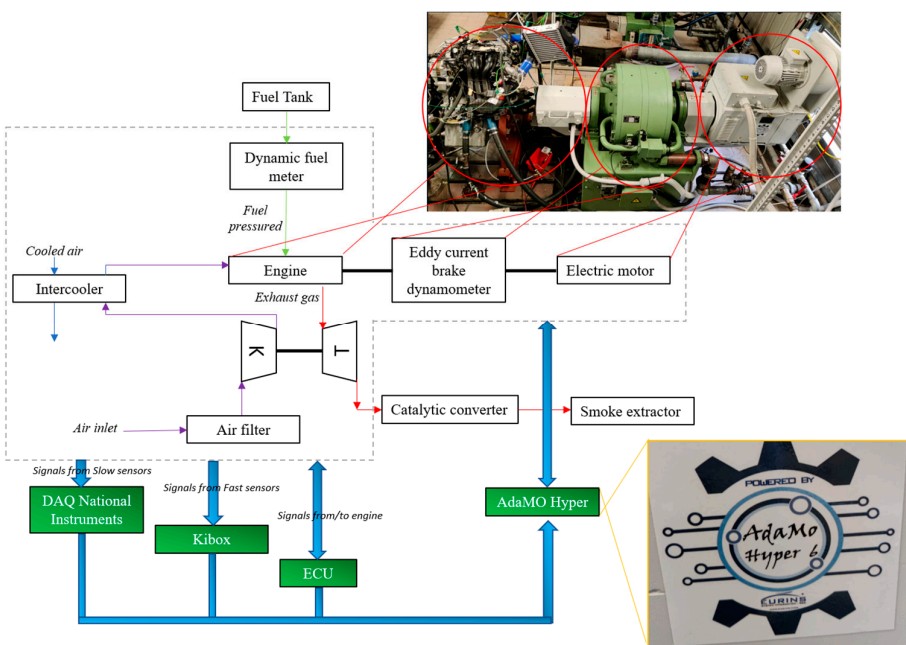

**Figure 1.** Experimental setup and acquisition system used to carry out the activities.

## 3. Artificial Neural Network Setup and Methods

The proposed method was fine-tuned, working on the dataset resulting from the experimental activities carried out on the three-cylinder PFI engine. The prediction of the delivered torque was performed via a NARX approach [26,27], i.e., a recurrent dynamic neural network used to model nonlinear dynamic systems and applied in time-series

modeling [28]. Such a network is composed of a series–parallel architecture (i.e., open-loop) or a parallel one (i.e., close-loop) (Figure 2). In the series–parallel architecture, the desired output value ŷ(t) is predicted from the present and past values of the input x(t) and the true past value of the time series y(t). In the parallel architecture, the prediction is performed from the present and past values of x(t) and the predicted value of ŷ(t). A series–parallel architecture is used during the training phase because of the availability of the true past value of the time series. Then, the architecture is converted into a parallel one, useful for multi-step-ahead forecasting.

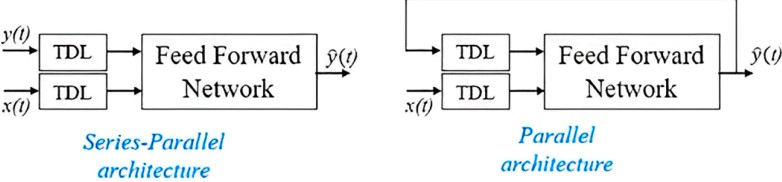

**Figure 2.** Architectures of the NARX neural network. TDL stands for tapped delay line, which delays an input by the specified number of sample periods and provides an output signal for each delay.

Five transient cycles (TC), each with 5760 samples, were realized through the AdaMo actuation, which caused the engine to run with variable engine speed and throttle valve opening. In such work, the engine speed varied from 1000 to 3500 rpm. The torque delivered by the engine continuously changed as a result (Figure 3). The target of the present activity was to predict such a parameter at the five conditions proposed.

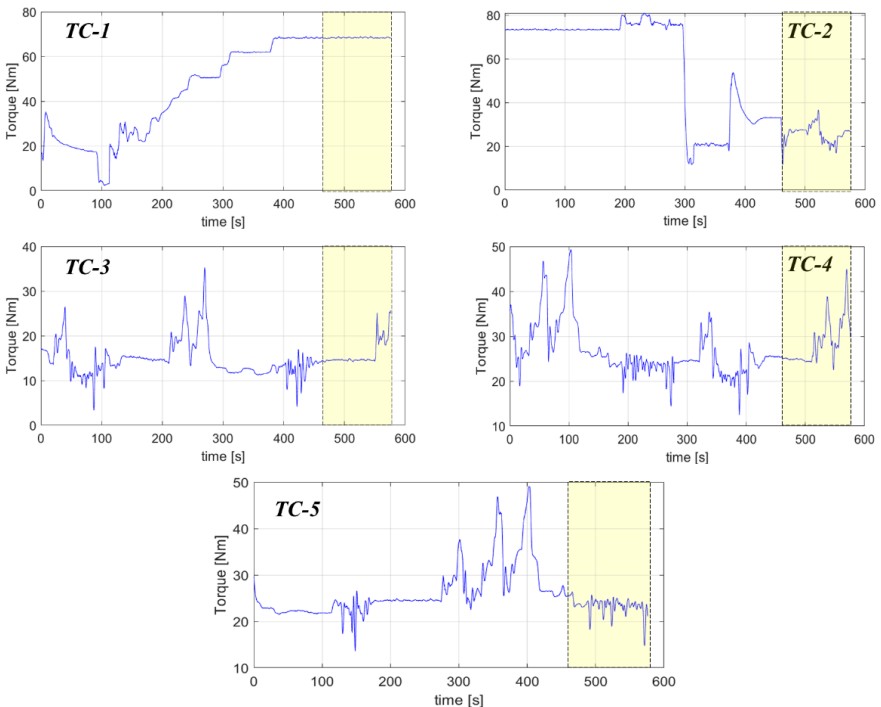

**Figure 3.** Delivered torque trend for the five tested datasets. The dotted boxes indicate the signals to be predicted as subsequently described.

Data from the ECU, pressure sensors, thermocouples, and torquemeter were acquired by AdaMo with a sampling frequency of 10 Hz. A total of 12 variables, considered among the most characteristics, were chosen as input to the neural structure to predict the torque delivered by the engine, namely:

- Pressure sensors and thermocouples:
  temperature of the air before the filter (TC_Air_Intake), temperature and pressure of the air at the intake pipe (TC_ETB_OUT and MAP), pressure and temperature of the exhaust gas before (TC_Turbine IN, P_Turbine IN) and after the turbine (TC_Turbine OUT and P_Turbine OUT), temperature of the engine oil (TC_Engine Oil).
- Engine control unit actuation:
  activation time of the injector (InjectionTime) and ignition timing of the spark (SparkAdvance) at the first cylinder beside the flywheel.
- AdaMo actuation:
  throttle valve opening (Throttle Position) and engine speed (Engine speed).

For each transient cycle analyzed, the entire dataset was composed of an input matrix of [12 × 5760] samples and an output matrix of [1 × 5760] (Figure 4a,b). For this application, 5760 samples represented a duration of approximately 120 s. This duration is similar to the length of the ECE-15 homologation cycle segment that involves the highest vehicle speeds. For the sake of completeness, Appendix A reports the input trend of the used variables for the TC-2 case.

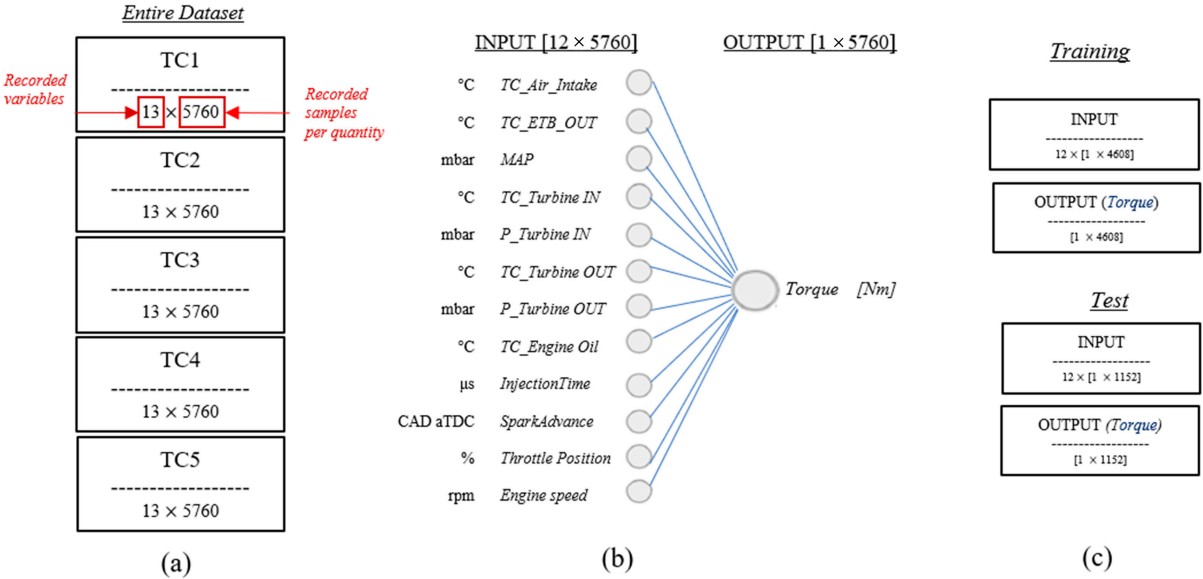

**Figure 4.** (**a**) Description of the entire dataset used in this activity; (**b**) division between input and output parameters for each case analyzed and displayed in (**a**); (**c**) dataset segmentation for the training and test session.

The definition of the neural structure in terms of number of neurons and hidden layers was performed through preliminary analysis. A random case among the five tested, i.e., TC2, Figure 3, was chosen for this purpose. According to the criteria analyzed in [29]:

- The number of hidden neurons should be between the size of the input layer and the size of the output layer.
- The number of hidden neurons should be 2/3 the size of the input layer, plus the size of the output layer.
- The number of hidden neurons should be less than twice the size of the input layer.

A maximum of 2 hidden layers composed of different numbers of neurons (9,12,15,18,21,23) (Figure 5) were tested, comparing the corresponding training performances. The structure showing the best performance in training was chosen for the test session.

A parallel preliminary analysis was also carried out by using the SHAP (Shapley additive explanation) method to evaluate the impact of the single measured quantities (Figure 4a) on the objective function (i.e., torque) [30]. SHAP aims to explain the prediction of an instance by calculating the contribution of each characteristic to the prediction [25].

The average absolute Shapley values (ABSV) [31] allowed the authors to quantify the impact of the single measured quantities on the objective function. The less influential variables were deleted from the initial dataset and the same analysis, previously executed with the entire dataset, was performed. Finally, the performance of the neural structure on the entire and reduced dataset was compared through the prediction of the delivered torque in the analyzed TC2. The structure providing the best results was employed for the torque prediction in the five analyzed cases. For each transient cycle, the training session was realized in the MATLAB environment on 80% of the entire dataset, while the test session regarding the torque prediction was performed with the remaining 20% (Figure 4c).

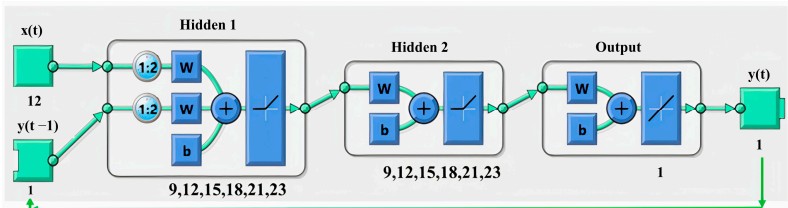

**Figure 5.** NARX structure used to perform the prediction of the torque delivered by the engine. A maximum of 2 hidden layers composed of different numbers of neurons (9,12,15,18,21,23) were selected to optimize the performance.

## 4. Results and Discussion

As reported in the previous section, the definition of the neural structure was performed through preliminary activities considering the transient cycle TC-2. Of the provided dataset, 80% was used for the training session. The training performance of 42 different combinations of neurons and hidden layers was evaluated in terms of RMSE [24]. As depicted in Figure 6, each combination shows an RMSE value under the acceptable threshold of 5% [32]. In particular, the structure composed of 2 hidden layers with 21 and 23 neurons respectively presented the best performance with an RMSE value equal to 3.37%. Such a structure, i.e., {21 23}, was selected for the next step, i.e., the test session, to predict the 20% of the remaining torque signal.

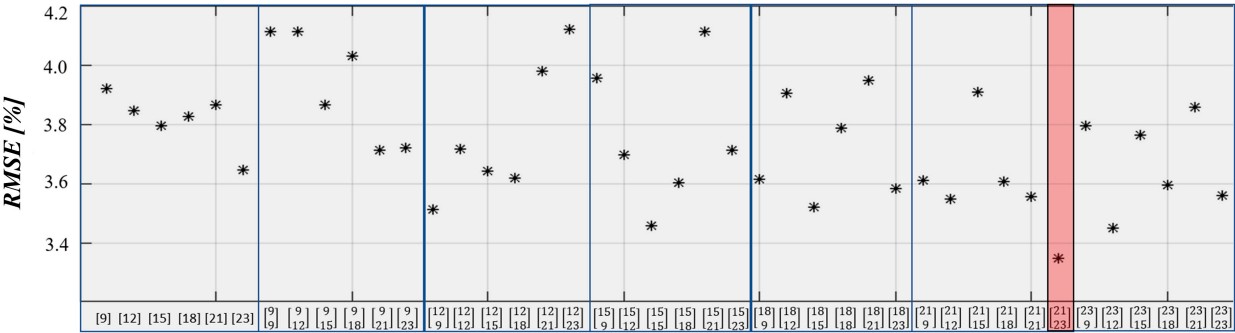

**Figure 6.** RMSE values for training of different combinations of neurons and hidden layers, based on the provided initial dataset. The red box indicates the combination chosen for the test session, i.e., the one composed by 2 hidden layers with 21 and 23 neurons, respectively.

Before testing the selected structure, the less influential input variables identified by the Shapley analysis on TC-2 were deleted from the initial dataset. Figure 7 shows the results of the sensitivity analysis carried out using the average absolute Shapley values. The temperature of the exhaust gas after the turbine, TC_Turbine OUT, the throttle position, and the engine SparkAdvance presented the lowest percentage of impact on the torque prediction.

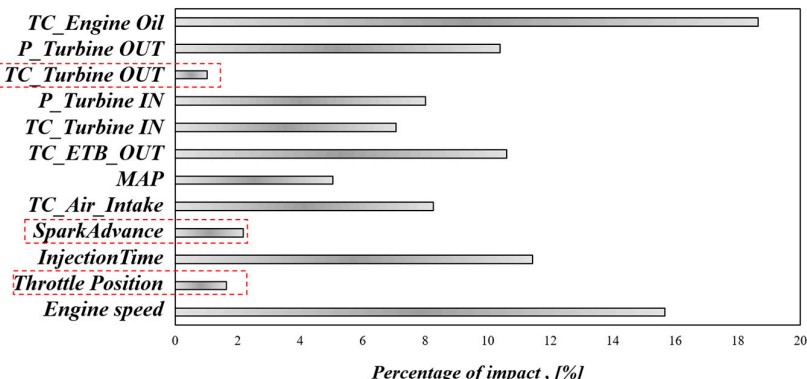

**Figure 7.** Shapley analysis: global interpretation of the feature's importance for the torque prediction.

By excluding such quantities, the number of inputs was reduced from 12 to 9. Following the considerations reported in the Section 3, a new neural structure optimization was required. Even in this case, a maximum of 2 hidden layers composed of different numbers of neurons (7,9,12,15,17) was selected following the architecture depicted in Figure 5, for a total of 25 combinations, for the corresponding evaluation of the performance during the training phase. The structure showing the best RMSE in training was chosen for the test session. As depicted in Figure 8, each combination showed an RMSE value under the acceptable threshold of 5%. In particular, it is worth mentioning that the combinations never exceeded 4% RMSE, unlike the cases analyzed by considering the entire provided dataset (Figure 6). This means that, on average, the learning ability of the neural structures improved when operating with the reduced dataset, together with the computational time. With the reduced dataset, the structure {17 15} performed the best, with RMSE = 3.21%, and for that reason was chosen to predict the torque signals.

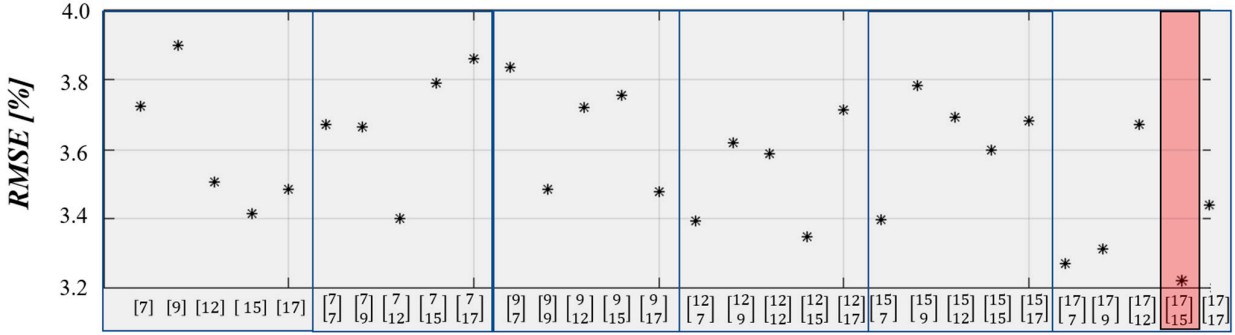

**Figure 8.** RMSE values for training of different combinations of neurons and hidden layers, based on the reduced dataset. The red box indicates the combination chosen for the test session.

Figure 9 displays the torque signals predicted by the two tested architectures, i.e., {21 23} for the entire dataset and {17 15} for the reduced one. For each forecast, the average deviation of the prediction from the target throughout the range was computed (Equation (1)):

$$\text{Err} = \frac{\sum_{i=1}^{n} \left[ \frac{|\text{Target}_i - \text{Predicted}_i|}{\text{Target}_i} \times 100 \right]}{n} \tag{1}$$

where n is the number of samples considered for the test case and i the ith sample. Several samples wrongly predicted equal to 10% were set to be considered as acceptable for the prediction. The average percentual error, i.e., $\text{Err}_{avg}$, was also computed to draw attention to the global prediction quality. For this kind of application, a maximum critical threshold of 10 is established for $\text{Err}_{avg}$.

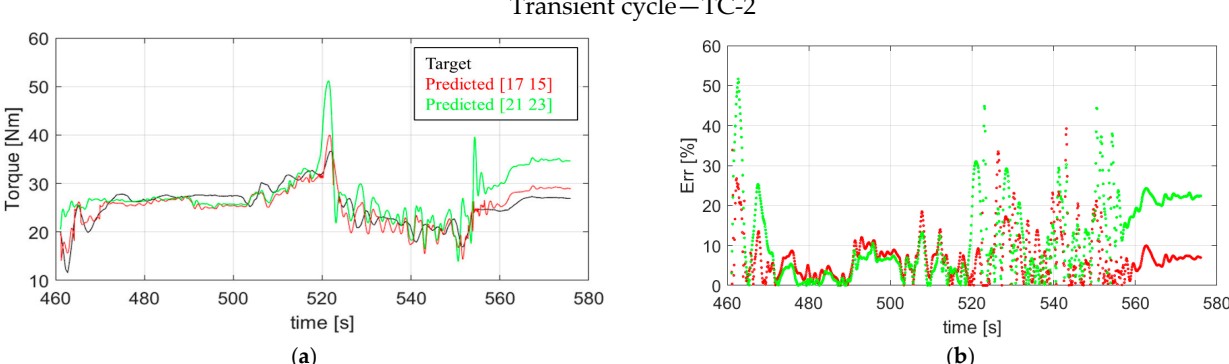

**Figure 9.** (**a**) Torque signals predicted by the NARX structure {21 23} (green curve) and {17 15} (red curve) and (**b**) corresponding %Err in forecasting the target (black curve in (**a**)).

From a qualitative standpoint, both models can reproduce the target trend. However, it is straightforward to observe how the structure trained on the entire dataset, i.e., {21 23}, shows larger amplitude fluctuations in the portions where the target tends to oscillate, particularly with overstated peaks in the regions of greatest torque, where the Err approaches 43%. On the contrary, the structure trained on the reduced dataset, i.e., {17 15}, manages to confine the peaks while remaining consistent with the trend of the target, presenting significantly smaller variations. Furthermore, at the extremes of the range analyzed, it exhibits a behavior that is much more consistent with the target than with the architecture {21 23}. The structure {17 15} has a maximum Err of 39% and an average of 7.01%, effectively placing it below the crucial area of 10%. In contrast, the $Err_{avg}$ of the structure {21 23} is equivalent to 11.44%, which is not acceptable. Considering the structure {17 15}, the biggest errors occurred in the area between 530 and 550 s, when the target signal oscillated around a torque value of roughly 25 to 15 Nm. At 39% inaccuracy, this amounts to an error of around 5 Nm. Given the nature of the target signal, such an error can still be considered a singularity, and in any event, inconsequential, especially when the average error value achieved is 7.97%. Furthermore, as shown in Figure 3, TC-2, the neural structure was trained on a small number of samples characterized by low variability, whereas the torque signal suddenly varied sample by sample during the test session. Therefore, such a condition further highlights the forecasting capabilities of the proposed structure. Based on these considerations, the structure {17 15} trained on the reduced dataset was definitively chosen for the prediction of the remaining four torque signals.

Figure 10 shows the forecasting results obtained by the neural structure {17 15} on the reduced dataset. Starting from the transient cycle TC-1, it is possible to highlight the opposite nature of such a dataset with respect to TC-2. TC-1 is characterized by high variability in the range utilized for the training sessions and by a torque signal almost constant in the test range. The structure had some issues tracking the frequent oscillations of the target signal, but it always managed to keep the forecast error Err below 1.2%. The average percentage error $Err_{avg}$ equals 0.43%, i.e., under the critical threshold of 10%, which testifies to the quality of the prediction. Moreover, the {17 15} forecast follows the target fluctuation across most of the analyzed range. The last three dynamic cycles examined in this work (i.e., TC-3, TC-4 and TC-5) are segments of transient cycles of the NEDC-EUDC type [33,34], which are characterized by a significant fluctuation of the quantities involved throughout the range of analysis (Figure 3). In terms of quality, the neural structure can track the oscillations of the target torque signal. In comparison to the TC-1 case, the average error values are higher, but still less than 10%: $Err_{avg}$ is equal to 1.64%, 3.03%, and 2.62% for TC-3, TC-4, and TC-5, respectively. For TC-3, the prediction error was always under 10%, while TC-4 and TC-5 showed singularities with Err over 10%. At TC-4, 48 predictions over 1150 target samples showed Err > 10% which represents about 4% of the total prediction. At TC-5, that value increased to 52, which represents 4.5% of the total samples to be predicted. Such results make the NARX prediction consistent in the cases analyzed.

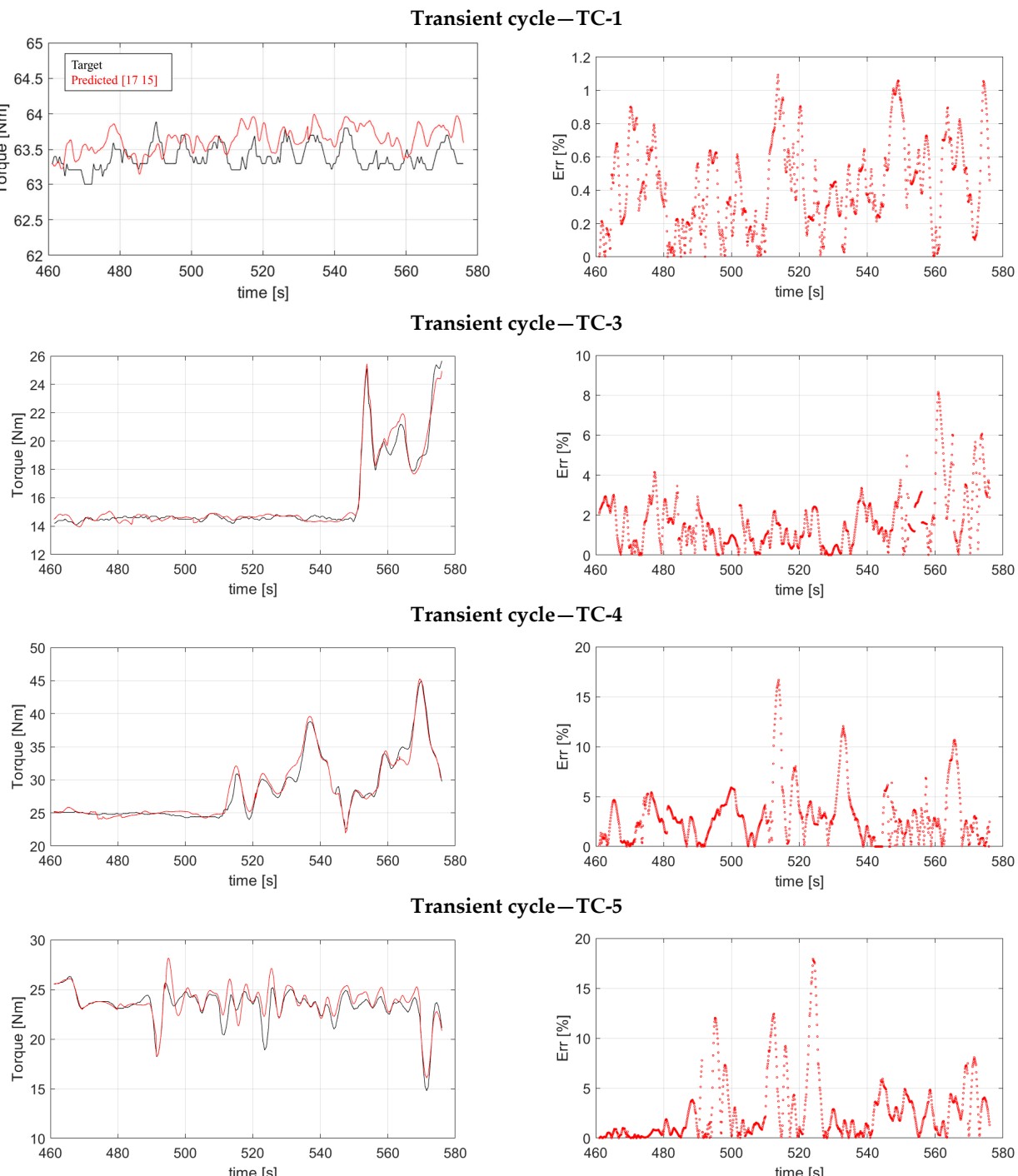

**Figure 10.** (left side) Comparison between the target torque signals (black curves) and the ones predicted by {17 15} (red curve) with (right side) the corresponding %Err.

As a final attempt, all of the trends were merged (TC-1 + TC-2 + TC-3 + TC-4 +TC-5) and a comparison was made between the prediction made by the structure {17 15} operating with the reduced dataset and the prediction made by the structure {21 23} operating with the complete dataset, to evaluate the quality of the performance when both were operating with longer dynamic cycles, i.e., characterized by a greater number of samples than those previously examined. As a result, the input dataset was a matrix of [9 × 28,800] values for the structure {17 15} and one of [12 × 28,800] values for the structure {21 23}, whereas the output dataset was [1 × 28,800]. The test specifications are shown in Figure 11. Of

the provided dataset, 80% was used for the training session and 20% for the test. In other words, the NARX structure was required to fully predict the TC-5 (Figure 3).

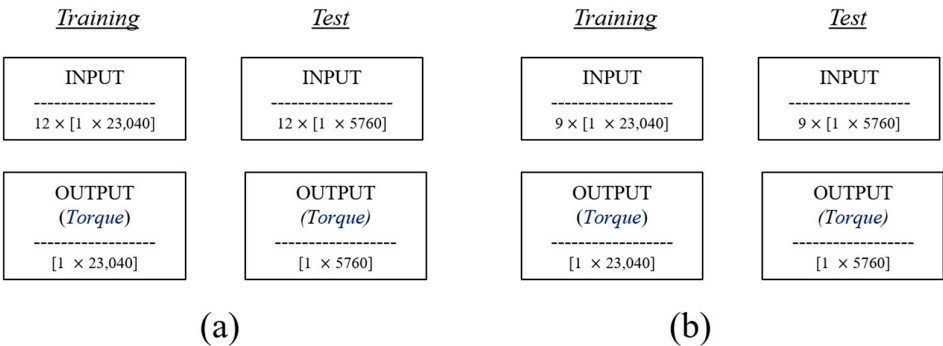

**Figure 11.** Specification of training and test session performed by (**a**) the NARX structure {21 23} on the entire dataset created by merging the transient cycles of Figure 3 and by (**b**) the NARX structure {17 15} on the dataset with a reduced number of inputs, according to the Shapley analysis.

Figure 12 shows the obtained findings of the forecasting activities carried out on the dataset of Figure 11. In general, both structures improved their performance, both qualitatively and quantitatively. From a qualitative point of view, the structures can perfectly reproduce the continuous changes of the target torque signals, differently from the other cases previously analyzed. This evidence is most likely attributable to the larger sample size expected for the training activity. From a quantitative point of view, both structures can guarantee an average error $Err_{avg}$ of less than 10%: {21 23} equals 1.13% and {17 15} of 0.99%. The structure {21 23} trained on the entire dataset made a number of predictions with Err greater than 10%, equal to 115 samples, and {17 15} did the same. Such a number corresponds to 2% of the total samples predicted. To sum up, even in this case, the structure {17 15} trained on the reduced dataset performed best.

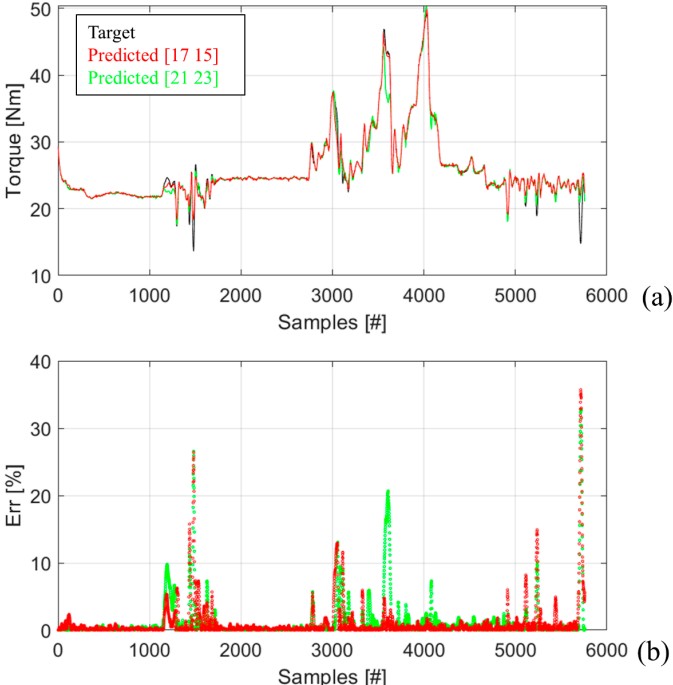

**Figure 12.** (**a**) Comparison between the target torque signals (black curves) and the ones predicted by {17 15} (green curve) and {21 23} (red curve) with (**b**) the corresponding %Err.

To validate the assumption that NARX can improve predictive performance when trained on a larger dataset, the 5 dynamic cycles proposed (Figure 3) were randomly merged in the following manner to produce new cycle dynamics represented by 288,000 samples: TC-2 + TC-5 + TC-1 + TC-3 + TC-2. The predictive performance of the structure {17 15} was evaluated and the findings are displayed in Figure 13. Even in these cases, 80% of the provided data (input = $1 \times [9 \times 23{,}040]$ and output = $[1 \times 23{,}040]$) were used for the training session and 20% for the test (input = $1 \times [9 \times 5760]$ and output = $[1 \times 5760]$). The structure {17 15} reproduced the target trend with $\text{Err}_{avg}$ of less than 10% and equal to 3.62%. Altogether, 223 samples, corresponding to about 4% of the total, were wrongly predicted by the structure. In particular, the maximum errors occurres close to the largest and most sudden changes in torque values. However, considering TC-2 (Figure 9), the architecture improved its forecasting performance.

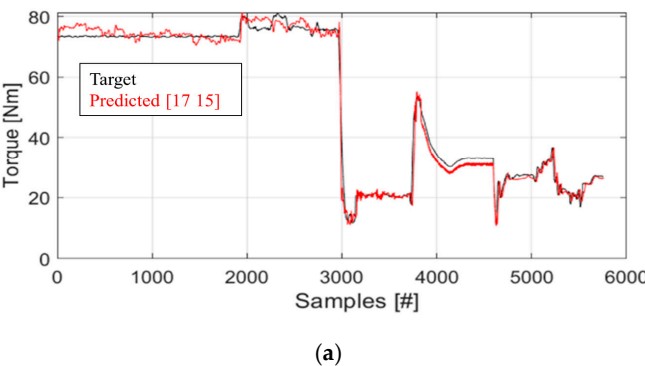

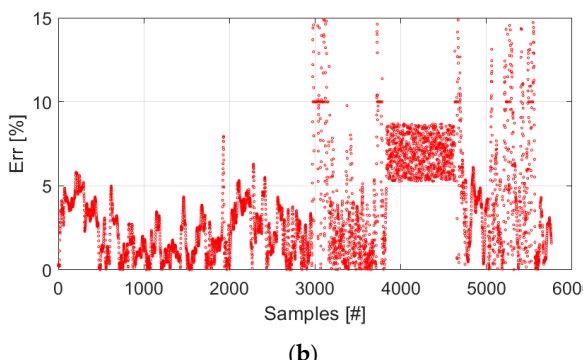

(**a**)                                                                 (**b**)

**Figure 13.** (**a**) Comparison between the target torque signals (black curves) and the ones predicted by {17 15} (red curve) with (**b**) the corresponding %Err.

## 5. Conclusions

The present work analyzes the performance of a nonlinear autoregressive with exogenous inputs (NARX) technique to predict the torque of an internal combustion engine. The neural architecture used was trained and tested on experimental data from physical sensors and the engine control unit under different operating conditions, on a port fuel injection three-cylinder spark-ignition engine. In a preliminary phase of the work, the optimization process of the NARX network used is presented. This activity is outlined in two fundamental steps: the first consists of the optimization of the internal structure of the network in terms of the number of hidden layers and the number of neurons per layer; the second consists of the Shapley sensitivity analysis aimed at evaluating the physical input quantities which most influence the target, i.e., the torque. The main findings are summarized below:

- The training performance of different combinations of neurons and hidden layers was evaluated in terms of RMSE on a specific case from the five analyzed in this work. All combinations showed RMSE values below the acceptable threshold of 5%. The structure with 2 hidden layers and 21 and 23 neurons, respectively, showed the best performance with an RMSE equal to 3.37%.
- The Shapley analysis performed on the entire dataset allowed identification of the least influential input variables for the prediction. These variables were excluded and therefore the number of inputs was reduced from 12 to 9.
- The NARX structure optimization performed on the reduced dataset showed the capability of the 25 combinations of neurons and hidden layers tested to achieve RMSE values below 5% during the training session. In particular, the structure with {17 15} neurons in 2 hidden layers showed the best performance with an RMSE of about 3%.
- The forecasting performance of the tested structures, i.e., {21 23} for the entire dataset and {17 15} for the reduced one, were evaluated on a specific case (TC-2). Both archi-

tectures reproduced the trend target; in particular, {17 15} showed smaller amplitude fluctuations and more consistent behavior with the target. An average error $Err_{avg}$ of about 7%, i.e., below the acceptable threshold of 10%, was shown by such a structure. Conversely, {21 23} generated $Err_{avg}$ of 11.44%, above the acceptable threshold.

- The structure {17 15} was evaluated on four other different cycles. It was able to follow the oscillations of the target signal, showing average errors always lower than 10%.
- The five cycles tested were merged and both structures, i.e., {21 23} for the entire dataset and {17 15} for the reduced one, performed better than the previous activities. The structure {17 15} showed $Err_{avg}$ of 0.99%, and {21 23} showed 1.13%.
- The five cycles were randomly merged and the forecasting performance of {17 15} was evaluated. Such an architecture showed $Err_{avg}$ of about 3.6% and an excellent ability to reproduce the target.

Several elements can be investigated in future study to improve the performance of the NARX technique for torque prediction in internal combustion engines. For instance, adding advanced optimization techniques and evaluating hyperparameter-optimizing approaches may enhance the training process. Furthermore, carrying out experiments with a wider and more diverse dataset, including various engine-operating conditions, may help in assessing the model's generalization capabilities. Investigating the potential of ensemble techniques or hybrid models integrating NARX with other prediction algorithms may result in improved accuracy and robustness with the aim to replace physical sensors in torque computation for internal combustion engines.

**Author Contributions:** Conceptualization, L.P. and F.R.; methodology, F.R. and C.N.G.; software, L.P.; validation, L.P. and F.R.; formal analysis, L.P. and F.R.; investigation, F.R.; resources, F.M. and C.N.G.; data curation, F.R.; writing—original draft preparation, F.R.; writing—review and editing, L.P.; visualization, L.P.; supervision, F.M.; project administration, F.M. All authors have read and agreed to the published version of the manuscript.

**Funding:** This research received no external funding.

**Data Availability Statement:** The data presented in this study are available from the corresponding author. The data are not publicly available due to privacy related choices.

**Conflicts of Interest:** The authors declare no conflict of interest.

## Nomenclature

| | |
|---|---|
| ERR | Percentage error |
| $ERR_{avg}$ | Average percentage error |
| ANN | Artificial neural network |
| ECU | Engine control unit |
| FFANN | Feed forward artificial neural network |
| HCCI | Homogeneous charge compression ignition |
| ICE | Internal combustion engine |
| ML | Machine learning |
| MLP | Multi-layer perceptron |
| MON | Motor octane number |
| NARX | Nonlinear autoregressive network with exogenous inputs |
| PFI | Port fuel injection |
| RBF | Radial basis function |
| RMSE | Root-mean-square error |
| RON | Research octane number |
| SI | Spark ignition |
| TDL | Tapped delay line |

## Appendix A

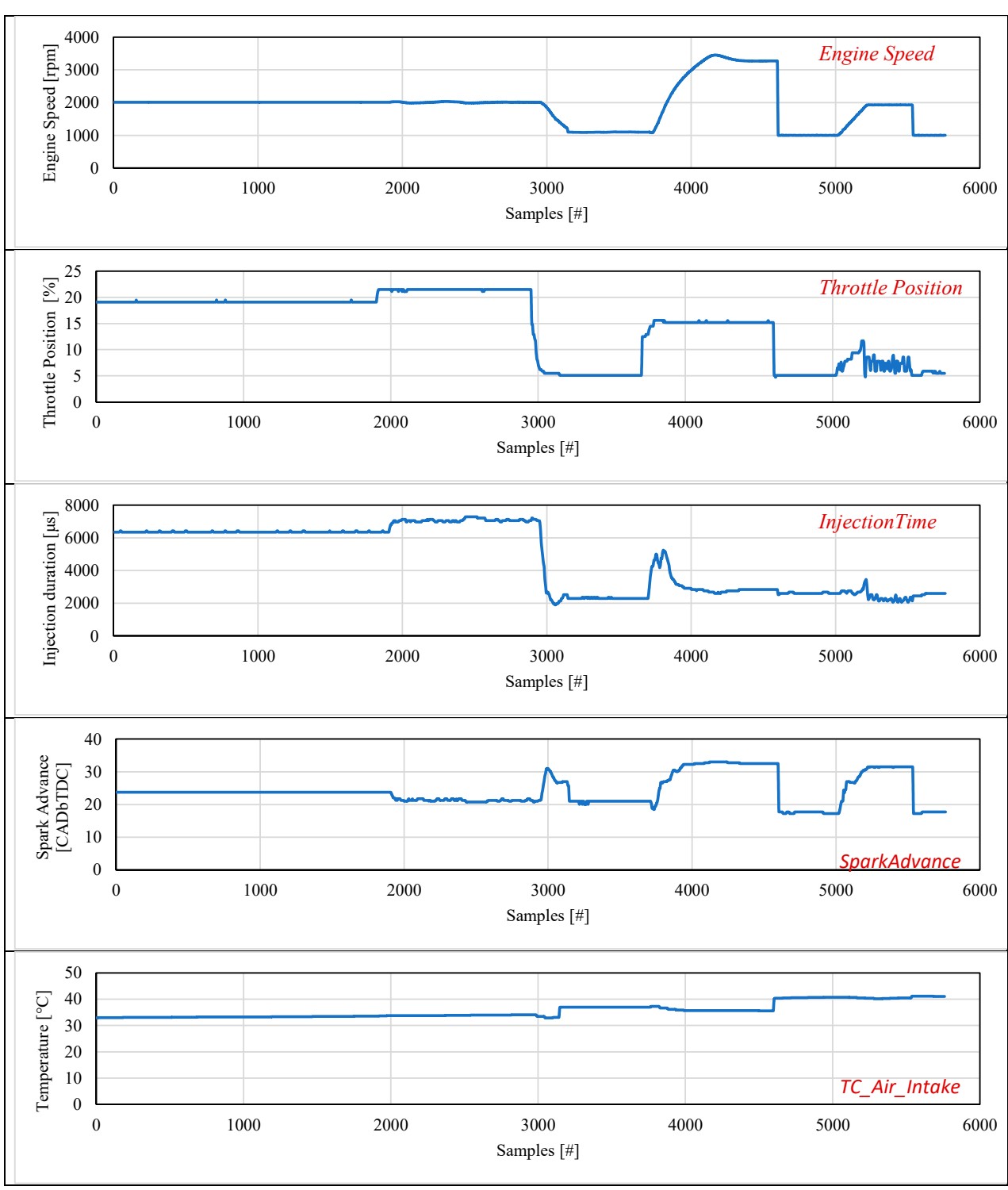

**Figure A1.** *Cont.*

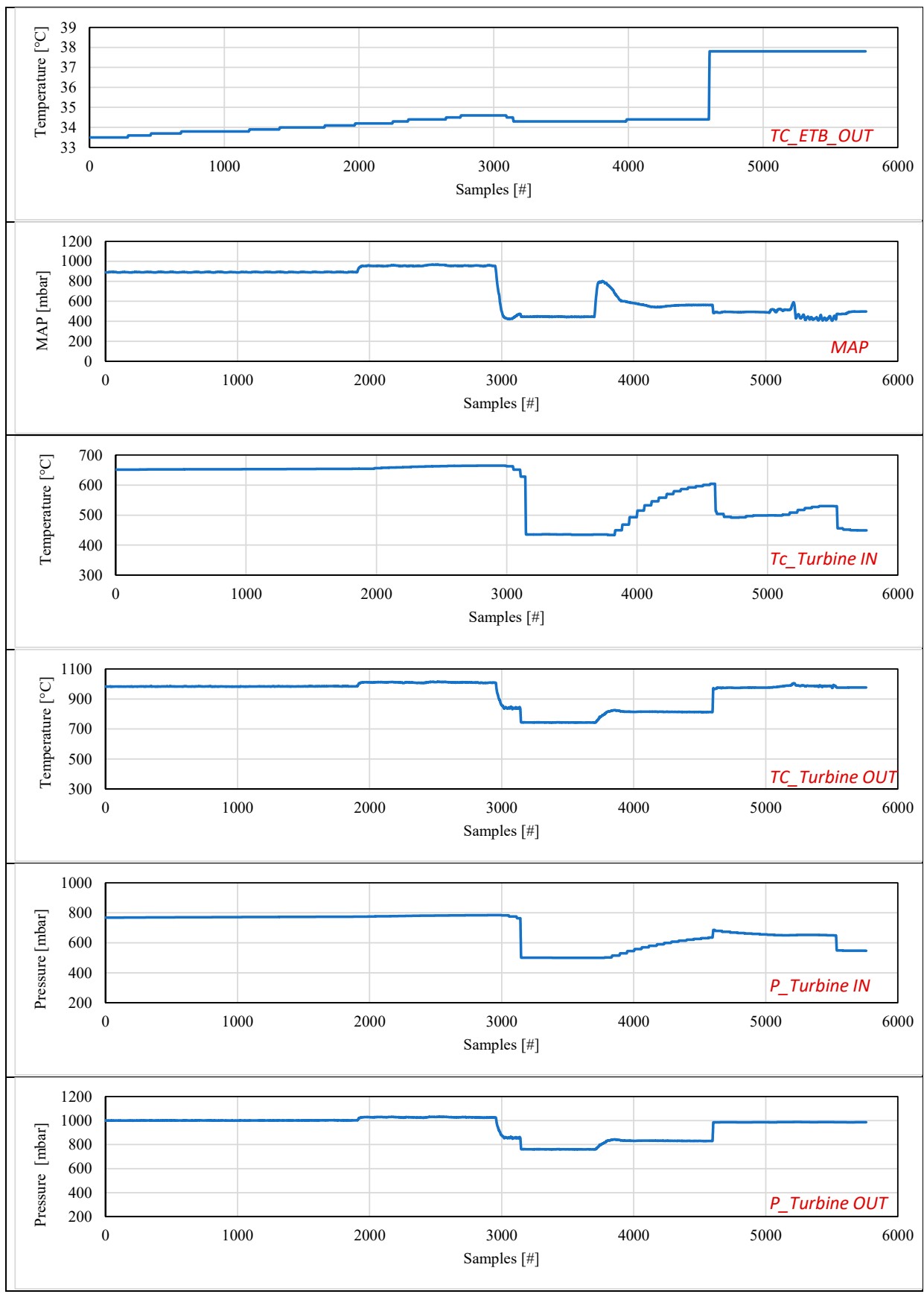

**Figure A1.** Experimental trend of the variables used in TC-2 as input and output to and from the tested NARX architecture.

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
