# Peer review of "NARX Technique to Predict Torque in Internal Combustion Engines"

_information, doi:10.3390/info14070417_

Round 1

Reviewer 1 Report

Dear Authors, please find my comments, questions, and suggestions below.

In lines 72 - 73, you wrote: "The signals coming from thermocouples TCK and pressure sensors PTX 1000 are acquired by data acquisition systems of National Instrument.". What National Instrument hardware and software did you use? Or did you only use the AdaMo Hyper software for data acquisition? You should note it in the article.

Please decipher the TDL abbreviation from Figure 2 in the text of the article. 

In lines 100 - 101, you wrote: "Five transient cycles (TC), each with 5760 samples, were realized through the AdaMo actuation, which causes the engine to run with variable engine speed and throttle valve opening.". Why did you choose exactly 5760 samples for the training phase? You should describe it in more detail. I believe this part of the study will be interesting and helpful to readers and your colleagues in the field.

In lines 122, 139, and 146, You refer to the whole of Figure 4. If you referred more precisely, as 4 (a), 4 (b), and 4 (c), then it would be more clear to readers. Please fix this.

Missed a word in a sentence in lines 122, 139, 146: "In particular, the structure composed of 2 hidden layers with 21 and 23 neurons respectively presents the best performance thanks to an RMSE value equal to 3.37%.". Please fix this. And it is not clear where the number 21 came from. It is not listed below the first hidden layer in Figure 5. Also, the sequences of the number of neurons in Figures 5 and 6 are different. You should eliminate this difference so as not to confuse the readers.

Also, I would recommend not writing combinations of numbers of neurons in square brackets in the text and in the Figures. This can be confusing because references to literature are given in square brackets. You could use curly braces instead.

In the conclusion section, you also could write about possible future works if it is planned.

In general, I think your article deserves publication in Information Journal after minor revision.

Author Response

Dear Revisor, you can find the responses in the attached file

Reviewer 2 Report

Title: "NARX technique for predicting torque in internal combustion engines" This scientific article is a highly innovative and consistent approach to the problem of torque prediction in internal combustion engines using the NARX (Nonlinear AutoRegressive with eXogenous inputs) technique. The authors undertake this task with extremely meticulous analysis, presenting results that are not only mathematically and statistically significant, but also have a significant impact on the automotive industry. The article focuses on the use of neural networks for torque estimation, which is a challenge due to the complexity of the system and the variety of factors affecting the result. The authors emphasize the growing need for more and more advanced computing systems that are able to manage the increasing number of variables in order to optimize internal combustion engines. Crucially, the paper highlights the growing role of "machine learning" in the automotive industry, especially in the context of prediction and optimization. The authors successfully use the NARX technique, performing preliminary activities aimed at optimizing the neural network in terms of the number of neurons and hidden layers, as well as the number of input parameters to be evaluated. Shapley's sensitivity analysis, used in the work, made it possible to assess the impact of individual variables on the predicted values, which allowed to reduce the amount of data processed by the architecture. Importantly, the optimized structure was able to achieve forecast errors at a level always below the critical threshold of 10%. The main conclusions that can be drawn from the article include that the structure with 2 hidden layers and 21 and 23 neurons, respectively, showed the best performance with an RMSE (Root Mean Square Error) of 3.37%. Shapley analysis across the entire dataset identified the least influential inputs on the forecast, which were then excluded, reducing the number of inputs from 12 to 9. An optimized NARX structure on a reduced dataset demonstrated the ability to achieve RMSE values ​​below 5% during a training session. In particular, the structure with 17 and 15 neurons in 2 hidden layers showed the best performance with an RMSE of around 3%. Moreover, this structure achieved an average error (Erravg) of around 7%, which is below the acceptable threshold of 10%. The article is not only highly informative, but also provides significant lessons for the automotive industry. It provides valuable information on how to make the best use of Machine Learning technology to optimize the operation of internal combustion engines. This work has the potential to significantly improve engine efficiency, which is crucial for the future of the automotive industry. The article is also impressive in terms of clarity and precision of data presentation. The use of different combinations of neurons and hidden layers, as well as Shapley's sensitivity analysis approach, are described in a clear and understandable way, which makes it easier to understand both for specialists and for readers with a less technical background. This paper is a significant contribution to the field of prediction and optimization in internal combustion engines, presenting the NARX technique as an effective variable management tool. This is definitely valuable reading for anyone with an interest in the field of Machine Learning or the automotive industry. The article presents many significant discoveries and conclusions. Nevertheless, there are several areas that may be further developed or explored in future work. Below are some suggestions: 1. Comparison with other machine learning methods: It would be useful to compare the NARX technique with other machine learning approaches such as SVM (Support Vector Machines) or decision tree-based methods. This would give the reader a broader view of the landscape of methods available and help understand where the NARX technique has its uniqueness strengths. 2. Studying the impact of noise and measurement errors: Often sensor data is polluted by noise or contains measurement errors. It would be useful to analyze the impact of these factors on the quality of forecasts. One could study how the NARX technique deals with data of varying degrees of purity. 3. Experiments with different neural network architectures: The authors focused on networks with two hidden layers. In future work, they could test how different configurations of neural networks (e.g. deeper networks, convolutional neural networks) affect the results. 4. Optimization of network hyperparameters: The work focuses on optimizing the number of neurons and hidden layers, but does not address other aspects of network learning, such as learning rate, regularization, or weight initialization strategies. These elements can significantly affect network performance and could be explored in future research. Recommendations: I would ask you to adapt the style of the article to the editorial requirements of MDPI. I would also ask for introduction of articles to the bibliography as a supplement to the topic. The work presents the method of validating the measurement results in an interesting way, in others the problem of machine learning as a problem in determining the credibility of the engine torque is presented. https://doi.org/10.1016/j.energy.2023.126974. https://doi.org/10.3390/s23094326, https://doi.org/10.1016/j.energy.2022.126002. https://doi.org/10.3390/info14040224, https://doi.org/10.1016/j.fuel.2023.128767

Author Response

(The authors gave the same response as above.)
